# Cell Wall Integrity Pathway Involved in Morphogenesis, Virulence and Antifungal Susceptibility in *Cryptococcus neoformans*

**DOI:** 10.3390/jof7100831

**Published:** 2021-10-05

**Authors:** Haroldo Cesar de Oliveira, Suelen Andreia Rossi, Irene García-Barbazán, Óscar Zaragoza, Nuria Trevijano-Contador

**Affiliations:** 1Instituto Carlos Chagas, Fundação Oswaldo Cruz (Fiocruz), Curitiba 81350-010, Brazil; haroldo.oliveira@fiocruz.br; 2Mycology Reference Laboratory, National Centre for Microbiology, Instituto de Salud Carlos III, Majadahonda, 28222 Madrid, Spain; su.arossi@gmail.com (S.A.R.); igarcia@isciii.es (I.G.-B.); ozaragoza@isciii.es (Ó.Z.); 3Department of Microbiology, Biomedical Sciences Institute, University of São Paulo, São Paulo 05508-000, Brazil

**Keywords:** cell wall, cell wall integrity (CWI) pathway, PKC, GTPases, MAP kinase, morphogenesis, virulence, pathogenesis

## Abstract

Due to its location, the fungal cell wall is the compartment that allows the interaction with the environment and/or the host, playing an important role during infection as well as in different biological functions such as cell morphology, cell permeability and protection against stress. All these processes involve the activation of signaling pathways within the cell. The cell wall integrity (CWI) pathway is the main route responsible for maintaining the functionality and proper structure of the cell wall. This pathway is highly conserved in the fungal kingdom and has been extensively characterized in *Saccharomyces cerevisiae.* However, there are still many unknown aspects of this pathway in the pathogenic fungi, such as *Cryptococcus neoformans*. This yeast is of particular interest because it is found in the environment, but can also behave as pathogen in multiple organisms, including vertebrates and invertebrates, so it has to adapt to multiple factors to survive in multiple niches. In this review, we summarize the components of the CWI pathway in *C. neoformans* as well as its involvement in different aspects such as virulence factors, morphological changes, and its role as target for antifungal therapies among others.

## 1. Introduction

*Cryptococcus neoformans* is a basidiomycetes yeast widely distributed in the environment that can behave as a pathogen in susceptible patients [1,2]. *C neoformans* infection is very common, healthy people with intact immunity are resistant to the infection. However, in those with altered immunity, this pathogen causes disease, principally cryptococcal meningoencephalitis. The most common predisposing condition for this disease is HIV/AIDS, although it also occurs in people with other types of immune impairment [3]. Based on a 2017 estimation, globally, there are around 215,000 cases and 180,000 deaths due to cryptococcal meningoencephalitis each year, most of which occur in Sub-Saharan Africa [4].

*Cryptococcus neoformans* has a cosmopolitan distribution [1] and has been isolated from multiple niches, mainly from pigeon guano and in swamps [5], so it can cause disease in a wide variety of organisms [6,7,8]. This implies that *Cryptococcus* has to adapt to many changes and different types of hosts. This is different from other fungi such as *Candida albicans* which practically only infects humans. *C. neoformans* is acquired by inhalation of spores that initially establish infection in the lungs [2,9,10], but under immunosuppressive conditions this pathogen can disseminate to the central nervous system (CNS). The capsule is the main phenotypic characteristic of *C. neoformans* and its principal virulence factor since it has a large number of effects on the host and the immune system [11,12,13]. Among others, the capsule inhibits phagocytosis, blocks the migration of lymphocytes and the production of antibodies [14,15,16,17]. The polysaccharide capsule is anchored to the outer layer of the cell wall [18,19,20] and this union needs to happen correctly since the cell wall in *C. neoformans* has an additional function that is to maintain the capsule [18,19]. There are different factors that affect the cell wall such as temperature or interaction with the host; therefore, the routes that regulate the structure of the wall are of great importance to understand virulence.

The cryptococcal cell wall is an essential structure composed mainly of glucans, chitin, chitosan and glycoproteins. All these components contribute to the cell wall rigidity and integrity [20,21,22,23]. The cell wall plays an important role in protecting the cell against different types of stress [21,22,23,24]. In addition, the cell wall mediates the interaction with the external environment through different receptors whose activation will trigger a different signaling cascade in the cell [20,21].

The cell wall integrity (CWI) pathway is the main route responsible for cell wall biogenesis and the maintenance of integrity in fungi [25,26]. This pathway was initially described as activated in case of “*cell wall problems”* [27,28]. This route has been well studied and characterized in the yeast *Saccharomyces cerevisiae* and is conserved among different fungal species [28,29]. Furthermore, this via is very similar to the PKC pathway in humans. The cellular integrity pathway is essential for the aging process, oxidative stress responses and cellular morphogenesis among others. This pathway is integrated by different receptors and factors that in a last step activate the MAPK (mitogen-activated protein kinase) module [27,28]. 

In this review, we will summarize the knowledge about the cell wall integrity pathway and its role in *C. neoformans*.

## 2. How Does the CWI Signaling Pathway Work in *C. neoformans*?

When the integrity of the cell wall is altered, there are specific surface receptors, although it is not known how these receptors detect stress. Once activated, they interact with the guanine nucleotide exchange factor (GEF) Rom2 that regulate the Rho1 GTPase, which is highly regulated by many subunits. In the case of *C. neoformans*, little is known about the cell surface sensors. Only a putative Mtl1/mid2 homologue encoded by the CNAG_03308 gene (Figure 1) has been identified but its role in the CWI signaling pathway has not been established [28]. However, three homologues of *S. cerevisiae* Rom2 have been found in *C. neoformans.* They are called Rom2, Rom20 and Rom21 [25,28]. The interaction of the membrane sensors with Rom2 activates the small GTPase Rho1, which then activates the protein kinase C (Pkc1). Three Rho GTPases have been identified in *C. neoformans*, Rho1, Rho10 and Rho11 [28,30]. Pkc1, the major kinase involved in cell integrity triggers the activation of a MAPK cascade. In *C. neoformans*, the MAPK module consists of three members; MAPKKK Bck1, MAPKK Mkk2 and MAPK Mpk1 [28]. Absence of these kinases results in alterations at cell wall, such as the content of chitin or chitosan [30,31,32], which highlights their importance in the maintenance of a proper cell wall structure. The differences and similarities of the CWI pathway in *S. cerevisiae* and *C. neoformans* are summarized in Appendix A.

The CWI pathway activation by several stress stimuli will result in the regulation of different transcription factors that will trigger an adaptive response. Classical studies on *S. cerevisiae* transcription factors activation through the CWI pathway revealed that the transcription factors ScRlm1 and the ScSwi4/ScSwi6 complex induce the expression of several genes related to cell wall biogenesis, [27,33]. Although not fully characterized, *C. neoformans* has homologues to ScRlm1, ScSwi4 and ScSwi6, named CNAG_03998, CNAG_07464 and CNAG_01438 respectively, that may play a role in the CWI pathway response and activation [27].

In addition, disruption of the transcription factor Crz1 in *C. neoformans* (homologue to calcineurin-responsive zinc finger ScCrz1 in *S. cerevisiae*), causes a phenotype similar to a CWI mutant (*cna1*), such as with increased susceptibility to cell wall stressors. This suggests that in *C. neoformans* Crz1 could be linked to CWI pathway [34,35], revealing CWI pathway as an open field of study in the cryptococcal research.

## 3. Cell Wall Integrity and Virulence Factors in *Cryptococcus neoformans*

*Cryptococcus neoformans* has developed and acquired several features that allow the infection and virulence in a wide of hosts, such as the polysaccharide capsule, growth at 37 °C, melanin production and morphological changes (capsule increase and Titan cells formation). Cell wall integrity is crucial to all of these processes. Responses to environmental stresses are mediated by MAPK phosphorylation cascade [31] and the components of the PKC1-MAPK signaling pathway are vital to maintain the integrity of the cell.

### 3.1. Adaptation at High Temperature

The ability of *C. neoformans* to grow at 37 °C turns it into an invasive pathogenic fungus and excellent model to study virulence, pathogenesis and host interaction. Most fungal species have the ability to grow optimally between 25–35 °C, however, only a few species of fungi are considered thermotolerant (growth > 35 °C and until 40 °C) [36].

The Pkc1 signaling pathway plays an important role in the response to thermal stress regulated by Rho GTPases [30]. In *C. neoformans*, three Rho GTPases (Rho 1, Rho10 and Rho11) have been identified that are important for cell growth and in response to temperature stress [25,30,31]. Ballou et al. suggested that CdC42 proteins perform complementary functions with other Rho-like GTPases in response to host and environmental temperatures [37]. Furthermore, recent studies have shown that Crz1 and Had1 transcription factors play an important role in cell wall integrity, thermotolerance and virulence in *C. neoformans* [38].

Besides, all mutants of MAPK module (*Bck1*, *Mkk2* and *Mpk1*) have a growth defect at host temperature and are consequently less virulent in mouse models [30,32]. 

### 3.2. Capsule and CWI

The capsule is the main phenotypic feature of *C. neoformans*. It is mainly composed of two polysaccharides, glucuronoxymannan (GXM), which constitutes 90–95%, and glucuronoxylomanogalactan (GXMGal) which is 5–8% [13,39]. It also contains a small proportion of mannoproteins (<1%) [39,40]. Despite its importance, the mechanisms involved in capsule synthesis are still not fully understood. The main model suggests that the main components are synthesized intracellularly in the endoplasmic reticulum (ER), and exported to the extracellular space in vesicles. Then, the content of these vesicles is released, and the polysaccharide fibbers attach to the cell wall, in particular, to the α-1,3-glucan [41,42]. At the moment, many genes have been involved in capsule synthesis. Furthermore, several transduction pathways are also required for this process. The main ones are the Pka1-AMPc pathway and other MAPKs such as Hog1. Relatively recent studies speak about the possibility that the MAPK *Bck1* module may have a role in the regulation of capsule synthesis [43].

### 3.3. CWI and Melanin

*Cryptococcus neoformans* has also the capacity to accumulate melanin, which is an insoluble dark pigment widely found in nature. Melanin synthesis in this fungus occurs only in the presence of dipholic compounds, such as L-DOPA and is dependent on the enzyme diphenol oxidase, which is encoded by two genes, *LAC1* and *LAC2,* with *LAC1* being the main producer of melanin [44]. These enzymes localize the melanin that accumulates in the cell wall and contributes to the survival of *C. neoformans*, maintaining the integrity of this structure [45]. Melanin protects fungal cells against stress factors, which facilitates the survival of this pathogen in the host [46,47,48,49,50]. Melanin also contributes to virulence [51] and dissemination from the lungs to other organs [48]. The synthesis of this pigment is dependent on activation of the Pkc1 pathway by diacylglycerol (DAG) [52]. Furthermore, the DAG C1 binding domain from Pkc1 is necessary for proper laccase localization at the cryptococcal cell wall and for melanin synthesis [31,53].

## 4. CWI Pathway and Morphological Changes in *Cryptococcus neoformans*

During infection, *C. neoformans* undergoes morphological changes that allow the fungus to evade and resist the host′s immune system [54,55,56]. *Cryptococcus* can increase its size in two different ways: by enlarging the size of the capsule without changing the size of the cell body, or by increasing both structures [57,58,59]. This last process results in the appearance of the so-called Titan cells [57,60], which can reach a size of up to 70 µm [58,60,61]. These changes have a great impact on the *C. neoformans* infectious process, as Titan cells are resistant to stress factors [60,62], they are not phagocytosed by host cells [60,63], and polarize the host immune response to a Th2 type response, a non-protective response against *Cryptococcus* infection [64].

Recent studies showed that some enzymes in the CWI pathway and in particular the Pkc1 are important in cryptococcal morphology [18,29,31,65]. In *Cryptococcus*, PKC can be activated by diacylglycerol (DAG) which is produced by the degradation of phospholipids [52] and by the Rho1 GTPase, activated by the Rom2 protein [18,25,31,35]. Cryptococcal PKC besides essential to CWI pathway activation and maintenance of cell integrity [31]. Pkc1 activation also plays an important role to the capsule and Titan cells formation, among other important virulence factors to *Cryptococcus*.

## 5. CWI Pathway and *Cryptococcus* Capsule Growth

Disruption of the *PKC1* gene in *C. neoformans* results in an aberrant capsule, and this phenotype is restored after complementation of the mutant strain with the wild type gene [31]. The effect on capsule size is not related to the production of the polysaccharide, which surprisingly is overproduced by *C. neoformans Pkc1* mutants, but to the polysaccharide attachment to the cell wall [31]. Heung et al. [52] described that deletion of the C1 domain of Pkc1, which prevents the activation of Pkc1 by DAG, leads to a significant decrease in capsule size with a significant decrease in the density of the polysaccharide density attached to the cryptococcal cells [52]. 

*Cryptococcus neoformans* mutants of different proteins from the Pkc1 cascade also revealed a strong role of this pathway in the integrity and maintenance of the cell wall, influencing cryptococcal morphology [25]. Ppg1 is a phosphatase, homolog to *S. cerevisiae* Sit4p that participates in the regulation of the Pkc1 pathway, and its disruption in *C. neoformans* showed it is important for cell integrity and results in a significant decrease in the capsule size, with a genotype resembling the *C. neoformans* capsule-deficient *cap59* strain [25]. A decrease in capsule size is also observed in a *C. neoformans* mutant to Lrg1, a GTPase-activating protein that facilitates the conversion of GTP to GDP on the Rho1p regulating the Pkc1 pathway [25]. 

Interestingly, a cryptococcal Pkc1-dependent phosphorylation transcription factor Sp1 appears to be important to regulate the expression of some cryptococcal virulence factors, including capsule. *C. neoformans sp1* mutant strain displays a significant increase in capsule size, even in non-inducing conditions, indicating that this transcription factor negatively regulates capsule formation in *C. neoformans* [35]. In contrast, a *sp1 pkc1* double mutant displays a hypocapsular phenotype, which correlated with a reduction of other enzymes involved in cell wall synthesis, such as the Fks1 β-glucan synthase. For this reason, the authors hypothesized that the decreased capsule size in this strain may be the result of a decrease in attachment sites to the capsular polysaccharide, impairing the correct capsule assembling [35]. 

Pkc1 plays an important role in connecting different cell wall integrity pathways. Pkc1-dependent Cck1 casein kinase I protein is a regulator of the dephosphorylation of Hog1 under stress conditions [66], being the Pkc1 pathway constitutively activated when *HOG1* is disrupted in *C. neoformans* [67] and this intimately connects the high-osmolarity glycerol response (HOG) and PKC pathways, essential for cell integrity. Although disruption of *CCK1* does not affect *C. neoformans* capsule, *CCK1* is also responsible for regulating phosphorylation of Mpk1, a *C. neoformans* MAPK essential to cell wall integrity [32]. However, the HOG pathway plays an important role in the encapsulation of cryptococcal cells [18], revealing the importance of cell wall pathways in capsule assembling.

The HOG pathway, one of the most important components of the MAPK cascade, regulates stress, sexual differentiation and virulence in *C. neoformans* through sensing external challenges, leading to protective responses against different stresses such as high osmolarity, ion concentration and temperature among others, ensuring cell integrity [32,67,68,69]. HOG pathway seems to play a role in the negative regulation of the cryptococcal capsule. Bahn et al. [67] showed that a *C. neoformans hog1* mutant strain presents an excessive growth in the capsule when compared to the wild type strain. Under normal conditions, Hog1 is phosphorylated by Pbs2 [32,67,70] and under stress, Hog1 is rapidly dephosphorylated. The phosphorylated status is important in the induction or repression of the capsule in *C. neoformans*, being the capsule repressed when Hog1 is phosphorylated [32], showing that the capsule is probably induced when cells are under stress when Hog1 is dephosphorylated. In contrast, Huang et al. [68] showed that deletion of Hog1 in the closely-related species *C. gattii* leads to a decrease in capsules size, which suggests that Hog1 function in virulence is strongly dependent on the genetic background or species from the *Cryptococcus* genus [68].

In *C. neoformans*, Hog1 regulates two transcriptional factors, Atf1 e Mbs1 [71,72]. Disruption of *ATF1* in *C. neoformans* leads a significant increase in capsule size, which resembles the phenotype of *hog1* mutants [72]. However, disruption of *MBS1* does not have any effect on capsule size, indicating that Hog1 under normal conditions represses capsule synthesis through Mbs1 [71]. Analysis of global gene transcription in a *C. neoformans hog1* mutant showed a modest increase of expression of capsule-associated (*CAP*) genes that does not explain this increase in capsule size, so the authors discussed that Hog1 regulates capsule size in *C. neoformans* through other non-characterized factors [73].

## 6. CWI Pathway and Titan Cells Formation

Titan cells formation is characterized by the enlargement not only of the capsule, but also of the cell body. The molecular characterization of Titan cells has been limited by the lack conditions to obtain them in vitro [74]. However, three independent groups have described several in vitro conditions in which *C. neoformans* significantly increases its cell size [65,75,76], which has facilitated to gain knowledge on the molecular mechanisms involved in Titan cells formation.

In vitro pharmacological inhibition of Pkc1 in *C. neoformans* impairs Titan cells formation [65], indicating that this pathway plays a role in cell growth in this fungus. Interestingly, induction of Titan cells in vitro occurs in the presence of serum, and it has been shown that the polar lipidic fraction of this component induces Titan cells formation in vitro. For this reason, it has been hypothesized that DAG present in the serum induces Pkc1 activation and Titan cells formation in vitro [65].

All these studies show that the morphological changes in *C. neoformans* are responses to extracellular environmental factors and that the proteins of the cell wall integrity pathways play a very important role in this process. Figure 1, brings an overview on cryptococcal CWI pathway and its impact in the *C. neoformans* morphological changes.

## 7. Cell Wall Integrity Signaling and Antifungal Therapies

The fungal cell wall is composed of molecules that are not present in the human body and since it is an essential structure for the viability, it constitutes an ideal target for antifungal compounds and immunotherapies [20]. The antifungal drugs currently used in the clinic are insufficient and limited and furthermore, some developing countries do not have access to all antifungals due to their high cost [77,78,79]. Some of the limitations of the study and approval of new antifungals are that it is a slow and expensive process. Moreover, selection of intrinsic and acquired resistance by some species of fungi is becoming a clinical problem, which raises the challenge of finding effective new therapeutic options [80,81].

Echinocandins (caspofungin, micafungin and anidulafungin) are a class of antifungals that inhibit the synthesis of β-glucan in the fungal cell wall through non-competitive inhibition of the enzyme β-1, 3-glucan synthase. This antifungal treatment is ineffective against *C. neoformans* because basiodiomycetes contain low amounts of β-glucan at the cell wall [20]. According to the literature, there are different studies on the effects of echinocandins on some components of the CWI pathway in *C. neoformans*. For example, mutants of the Pkc1 pathway of *S. cerevisiae* are more sensitive to caspofungin [82,83]. However, Gerik et al. described that there were no significant differences between mutants of the components of the Pkc1 pathway and wild strains exposed to caspofungin in *C. neoformans* [25]. The authors refer to the fact that there are some differences between the species, since *C. neoformans* has a single MAP kinase kinase (MKK2) involved in the phosphorylation cascade of Pkc1, in contrast to *S. cerevisiae*, where there are two functional ones, Mkk1 and Mkk2 [25]. The difference in resistance to caspofungin between these two fungi may be due to the fact that in *S. cerevisiae* there are many genes of the Pkc1 pathway that are missing, while this has not been observed in any mutant of *C. neoformans* in susceptibility to caspofungin [25,82]. 

Exposure to caspofungin in *S. cerevisiae* leads to rapid activation of the Pkc1 pathway through *SLG1*, which is a surface sensor that has no homologue in *C. neoformans* [82].

It has also been described that *MPK1* is essential for the survival of *C. neoformans* in the presence of caspofungin. There is a relationship between calcium/calmodulin-dependent protein phosphatase calcineurin and *MPK1* in regulating cell integrity, since in the absence of functional calcineurin, *MPK1* helps to protect against caspofungin. Furthermore, mutants lacking *MPK1*1 are more sensitive to caspofungin compared with the wild type strains [32]. 

In *C. neoformans*, there is a relationship between calcineurin and the CWI signaling pathway at the transcriptional level. The *crz1* mutant has other phenotypes associated with CWI, as sensitivity to caspofungin. *CRZ1* in *S. cerevisiae* is associated with calcineurin and may play an important role in CWI signaling pathway [28,35]. 

Nowadays the treatment used for cryptococcosis is Amphotericin B (AmB), Fluconazole (FLZ) and 5-flucytosine (5FC). In severe cases, a combination of AmB and 5FC is applied and FLZ is used as maintenance treatment [84]. However, the treatment used with AmB and 5FC has shown some difficulties since in developing countries, 5FC is not available and the toxicity of the treatment persists. In regions with limited resources, the treatment available is often only with FLZ [85]. 

Fluconazole is a triazole antifungal and acts on the enzyme P450 lanosterol 14α- demethylase (ERG11), inhibiting the synthesis of ergosterol. Prolonged administration of FLZ can result in failure of the treatment of cryptococcosis and induce relapses in the patient. Besides, Sionor et al. in 2009 identified in clinical isolates of *C. neoformans* a pattern of intrinsic resistance to FLZ, called heteroresistance [86]. PKC-MAPK cascade has been demonstrated to affect the susceptibility to FLZ in *C. albicans* and *S. cerevisiae* [87,88]. Lee et al. suggested that in *C. neoformans* PKC-MAPK pathway is also involved in susceptibility to FLZ. The study analyzed the effects of deletions of the components of the Pkc1 pathway, Mkk2, Bck1 and Mpk1 pathway in *C. neoformans* mutants and they observed that these mutants showed hypersensitivity to FLZ compared to wild type strains [89].

In a recent work published by Sellers-Moya et al. the authors showed how imidazoles, and particularly clotrimazole, trigger more changes in MAPK phosphorylation than triazoles, which implies an activation of MAPKs Hog1 and Slt2 high HOG and CWI pathways, respectively in *S. cerevisiae* [90]. They showed that clotrimazole induces a high level of ROS in this yeast compared with fluconazole. In addition, resistance to this drug depends on both routes, HOG and CWI pathways. Regarding CWI they observed the activation by clotrimazole of Slt2 does not depend on pkc1 [90]. Amphotericin B alters the permeability of the fungal membrane by forming ion channels. It has been described that mutants with defects in cell integrity demonstrated increased sensitivity to AmB [91]. In particular, in a work carried out with different strains of *C. neoformans*, the *bck1* mutant (absence of MAPKKK in the cell integrity MAP kinase cascade) was more sensitive to AmB compared to wild type strains [91].

As mentioned above, the components of the fungal cell wall are not present in humans, so this structure is an excellent target for antifungal therapy. The CWI pathway is required for the adaptation of all types of stresses that disturb the cell wall, among which are included antifungals [28]. Although humans have homologous targets to the core components of the CWI pathway (Pkc1, MAP kinases, or Rho GTPases), studies in this pathway can optimize the specificity of the inhibitory components as will be described below.

Compounds that have a mechanism of action targeting the inhibition of CWI signaling pathway appear as promising options for the development of new antifungals, since it has been shown that *mpk1* mutants, which do not have functional CWI, are avirulent in murine models of cryptococcosis and do not survive human body temperature [32,92]. 

To find molecules with antifungal action against *C. neoformans*, Hartland et al. performed a screening with more than 360,000 molecules. As a result, the authors highlight molecules that can interfere with the fungal cell wall integrity, impacting the signaling of the cell wall integrity MAP kinase cascade [92].

## 8. Perspectives

The fungal cell wall is an essential structure for the cell because it maintains the cellular integrity and viability but also mediates the connections of the cell with the environment. The cell wall protects cells against stress conditions from the environment or the host. Cells respond and adapt to stress through a complex transcriptional program regulated by signaling pathways. The cell wall integrity pathway is the key for regulating this adaptive response. The structure of the cell wall and CWI is conserved among fungi, however, in the case of *C. neoformans* there are many aspects of this pathway that are still unknown. In this review, we have summarized how this pathway is involved in different aspects of the biology, including those involved in virulence (polysaccharide capsule and melanin). Several MAPK kinase pathways (such as Hog1 or Pkc1) are important in these processes. In addition, one of the most striking characteristics of *C. neoformans* is the ability to significantly increase its size (appearance of Titan cells), and some enzymes from the CWI pathway are important in these morphological changes. Finally, the use of the CWI pathway as a pharmacological target can help to improve existing antifungals as well as to develop new and more effective treatments. Therefore, the CWI pathway is primordial for the intrinsic adaptation to stress by temperature or antifungals and is crucial for virulence factors of *C. neoformans*.

## Figures and Tables

**Figure 1 jof-07-00831-f001:**
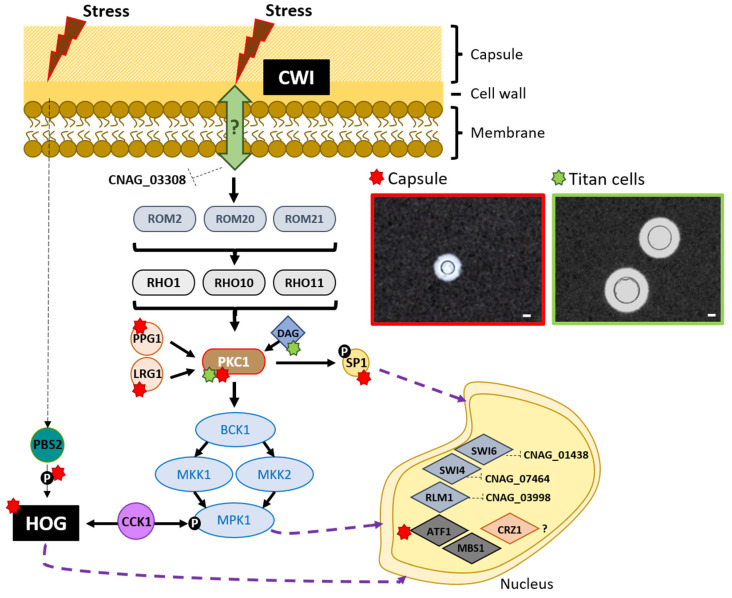
Cryptococcal CWI pathway and its impact on *Cryptococcus* morphological changes. The complete map of the cryptococcal CWI pathway is still under construction. In *Cryptococcus*, the plasma membrane are sensors that capture the environmental stress stimuli is not known, being just identified a homologue to Mid2, that is a transmembrane CWI pathway receptor in *S. cerevisiae*, named CNAG_03308, however, its role in CWI pathway is still unknown. The CWI pathway follows through the interaction with guanine nucleotide exchange factor (GEF), that *Cryptococcus* presents three homologues Rom2, Rom20 and Rom21, that may activate small GTPases Rho1, Rho10 and Rho11, which activates Pkc1, that is regulated by Ppg1 and Lgr1. Besides the small GTPases, DAG is also able to activate Pkc1. Pkc1 will be responsible to continue the pathway through regulation of the MAPK (mitogen-activated protein kinase) cascade (Bck1, Mkk1, Mkk2 and Mpk1) essential to the cell wall maintenance. Besides this, Pkc1 is also important to the phosphorylation of the cryptococcal transcriptional factor Sp1, important to regulate virulence factors expression in *Cryptococcus*. At the end of the pathway, several transcription factors (represented in the nucleus) will control the expression of factors that will work on the maintenance of the cell wall integrity. In *Cryptococcus*, some integrants of the CWI pathway are important for morphological changes that cryptococcal cells suffer during the interaction with host: capsule (red signal) and Titan cells (green signal) formation. Some factors such as Cck1 connect CWI pathway to other cell wall integrity pathway HOG, whose factors also contribute to morphological changes in *Cryptococcus*. In the figure, the bars represent 20 µm. This figure was elaborated based on [18,27,28].

## Data Availability

Not applicable.

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
