# Peer review of "Cell Wall Integrity Pathway Involved in Morphogenesis, Virulence and Antifungal Susceptibility in Cryptococcus neoformans"

_jof, 2021, doi:10.3390/jof7100831_

Round 1
Reviewer 1 Report
This is a comprehensive review linking CWI pathway with different virulence factors in Cryptococcus neoformans. However the manuscript would be improved by proofreading the English. In particular, there are some sections that are a bit confusing.
For example lines 92-95 - 'similar to CWI mutant' - what CWI mutant do you mean 'a' CWI mutant. Why is the phrase "CWI pathway as an open field" used after you are referring to CRZ1? Or is this a summation statement for the entire section.
Lines 161-165 need to be reworded and there are other sections too that could be improved.
Line 275 - 284 the discussion of synergy of targeting pathways and increased sensitivity to echinocandins is unclear. It states that the lack of synergy (as seen in S.c) may be due to only 1 (vs 2) Mkk2 gene - why? Why couldn't it just be due to the lack of B 1-3 glucan in the cell wall.
Figure 1 - thank you for this figure. Is HO = HOG?
Author Response
We thank the reviewer for all his comments and we are sorry that he/she has found certain sections confusing and with errors in the English. We hope that with the changes we have made it will be easier to understand.
Point 1. For example lines 92-95 - 'similar to CWI mutant' - what CWI mutant do you mean 'a' CWI mutant. Why is the phrase "CWI pathway as an open field" used after you are referring to CRZ1? Or is this a summation statement for the entire section.
Response 1. In lines 92-95 we refer to a CWI mutant. This error in this sentence has been corrected, now is “ In addition, disruption of the transcription factor Crz1 in C. neoformans (homologue to calcineurin-responsive zinc finger ScCrz1 in S. cerevisiae), causes a phenotype similar to a CWI mutant (cna1), such as with increased susceptibility to cell wall stressors.
We have also put the name of the mutant in parentheses (cna1).
Regarding the sentece. “CWI pathway as an open field of study in the cryptococcal research”. We want to say that the CWI pathway is a field that still needs to be studied in Cryptococcus because there are many aspects that are still unknown and it can be very interesting
Point 2. Lines 161-165 need to be reworded and there are other sections too that could be improved.
Response 2: The paragraph on lines 161-165 has been rewritten. Now is “Is now " Cryptococcal PKC besides essential to CWI pathway activation and maintenance of cell integrity. Pkc1 activation also plays an important role to the capsule and Titan cells formation, among other important virulence factors to Cryptococcus”. In addition, we have made a careful reading of the entire manuscript and other sections have been modified.
Point 3. Line 275 - 284 the discussion of synergy of targeting pathways and increased sensitivity to echinocandins is unclear. It states that the lack of synergy (as seen in S.c) may be due to only 1 (vs 2) Mkk2 gene - why? Why couldn't it just be due to the lack of B 1-3 glucan in the cell wall.
Response 3: In paragraph 278-282, we discussed that echinocandins are a type of treatment that is ineffective against Cryptococcus for this reason, it contains less amount of B-glucans. Nevertheless, in the work carried out by Geril et al., with different mutants, they see that in Cryptococcus there are no differences between the mutants of the PCK1 pathway and the wild type strain when exposed to Caspofungin and they conclude that this may be due to that Cryptococcus have a single MAP kinase while S. cerevisiae has two.
In this review, we have tried to summarize everything that is known about the CWI pathway in Cryptococcus and we have found it interesting to name all the related works, such as that of Gerik et al.
Point 4. Figure 1 - thank you for this figure. Is HO = HOG?
Response 4. The error in figure 1 has been corrected. It is HOG

Reviewer 2 Report
In the review article “Cell wall integrity pathway involved in morphogenesis, virulence and antifungal susceptibility in Cryptococcus neoformans” authors summarized the currently available information on signaling pathways related to the cell wall of this pathogen. It is nice work and valuable contribution to the field, giving access to interesting information, but the manuscript requires some improvements to make it more reader-friendly. My suggestions are as follows:
- In lines 27-28 authors wrote that “Cryptococcus neoformans infects most people, but only a few develop disease, principally cryptococcal meningoencephalitis…” I suggest changing this sentence and clarifying it, as the infection process is usually associated with the appearance of disease symptoms; it was mentioned about contact or exposure?
- Figure 1 should be placed within the text much earlier, i.e. before point 2. In the Figure 1, some graphic elements and included gene and protein abbreviations should be enlarged because they are unreadable, font sizes within the same level should be the same (i.e. Rom2, Rom20, Rom21). Shouldn't there be HOG in the figure instead of HO?
- In the text in points 2-4 there are many repetitions of particular words (i.e. lines 41-42 main, lines 58-60 this route, lines 151-152 capsule and cell body, lines 142-143 melanin). This part of the manuscript needs to be re-edited stylistically to avoid repetition not only of parts of sentences but also of entire fragments in order to make the text more coherent, legible and easy to follow.
- In point 2 authors compare C. neoformans and S. cerevisiae, but at the beginning of the text it is unclear which species the description refers to. This part of the text needs to be clarified. Perhaps this is where a table would work, in which authors would summarize the similarities and differences between the two species in relation to CWI signaling pathways.
- Throughout the text, it should be checked that the names of the species appear in italics. It should be also verified the correct writing of genes and proteins names and abbreviations (italics, uppercase or lowercase letters) - there are several mistakes in the text.
- Missing spaces, parentheses, extra spaces, e.g. line 44, 69 ... should be corrected throughout the text.
- Only the first time the species name should be written in full, then C. neoformans should be used.
- Incomplete sentence in line 139.
- The abbreviation TC appears only once at the end of the manuscript.
Author Response
We thank the reviewer for all his comments to improve this manuscript and we are happy that she/he thinks it is a good contribution to the field. We have tried to make all the changes proposed by the reviewer, our responses are as follows:
- Point 1. In lines 27-28 authors wrote that “Cryptococcus neoformansinfects most people, but only a few develop disease, principally cryptococcal meningoencephalitis…” I suggest changing this sentence and clarifying it, as the infection process is usually associated with the appearance of disease symptoms; it was mentioned about contact or exposure?
Response 1. In this paragraph what we want to highlight that Cryptococcus infects everyone but in healthy people it does not cause disease and the pathogen is eliminated, only those people with some type of immunosuppression Cryptococcus can cause disease. We wanted to say it in a different way since the introduction of most articles of Cryptococcus neoformans are very similar. On lines 36-40, we explain it in more detail and we believe that it is well understood how is the infection of this pathogen.
- Point 2. Figure 1 should be placed within the text much earlier, i.e. before point 2. In the Figure 1, some graphic elements and included gene and protein abbreviations should be enlarged because they are unreadable, font sizes within the same level should be the same (i.e. Rom2, Rom20, Rom21). Shouldn't there be HOG in the figure instead of HO?
Response 2. Thank you very much for this observation. The truth is that we have not specified where figure 1 should go. But in this new version we have proposed to the editors to put figure 1 after point 2, where we talk about the CWI in Cryptococcus. The typographical errors in figure 1 have already been corrected. And We regret this error in figure 1, it is HOG and not HO. This error has already been corrected.
- Point 3. In the text in points 2-4 there are many repetitions of particular words (i.e. lines 41-42 main, lines 58-60 this route, lines 151-152 capsule and cell body, lines 142-143 melanin). This part of the manuscript needs to be re-edited stylistically to avoid repetition not only of parts of sentences but also of entire fragments in order to make the text more coherent, legible and easy to follow.
Response 3. We regret the repetition of certain words throughout the manuscript. All those mentioned by the reviewer have been modified and we have also made a careful reading of the text and some parts have been rewritten to improve them and make them easier to understand.
- Point 4. In point 2 authors compare neoformansand S. cerevisiae, but at the beginning of the text it is unclear which species the description refers to. This part of the text needs to be clarified. Perhaps this is where a table would work, in which authors would summarize the similarities and differences between the two species in relation to CWI signaling pathways.
Response 4. All section 2 refers to CWI pathway in Cryptococcus neoformans (as specified in the title of that section). When we want to point out something about Saccharomyces cerevisiae it is specified in the text but even so we have made a table to make it easier for the reader to understand.
- Point 5. Throughout the text, it should be checked that the names of the species appear in italics. It should be also verified the correct writing of genes and proteins names and abbreviations (italics, uppercase or lowercase letters) - there are several mistakes in the text.
Response 5. We have corrected all the errors in the text regarding the nomenclature of genes, proteins and mutants. We are very sorry that the first version of this manuscript contained this type of error. But the truth is that we have named them as they appeared in the original articles that are referenced here
- Point 6. Missing spaces, parentheses, extra spaces, e.g. line 44, 69 ... should be corrected throughout the text.
Response 6. As we have commented in previous points of this letter, we have had a careful reading of this new version of this review and we have corrected all these errors. We have paid special attention to the extra spaces, lack of parentheses and other types of errors to improve the text.
- Point 7. Only the first time the species name should be written in full, then neoformansshould be used.
Response 7. Regarding this reviewer's comment, we have to say that we have used Cryptococcus neoformans the first time it is mentioned in the text and in all the beginning of the paragraph (since an abbreviation should not be used when starting a new paragraph). The rest of the times the abbreviation C. neofomans has been used. Anyway we have read all the text and modified those cases in which the abbreviation was not used We hope that the reviewer does not mind that we keep this style in the text.
- Point 8. Incomplete sentence in line 139.
Response 8. We regret this error that has already been corrected in the text. The sentence on line 139 is now “ These enzymes localize the melanin that accumulates in the cell Wall and contributes t the survival of C. neoformans, maintaining the integrity”.
- Point 9. The abbreviation TC appears only once at the end of the manuscript.
Response 9. Throughout the text we talk about Titan cells, this abbreviation TC at the end of the text has been an bug error we have not seen and it has already been corrected in the new version. We regret this mistake.

Round 2
Reviewer 2 Report
Authors have taken into account most of the suggestions and introduced changes to the manuscript, however I still have a few comments.
Response 1. In this paragraph what we want to highlight that Cryptococcus infects everyone but in healthy people it does not cause disease and the pathogen is eliminated, only those people with some type of immunosuppression Cryptococcus can cause disease. We wanted to say it in a different way since the introduction of most articles of Cryptococcus neoformans are very similar. On lines 36-40, we explain it in more detail and we believe that it is well understood how is the infection of this pathogen.
Thank you very much for the clarification, but I still think that the word "infect" should be somewhat mitigated as it suggests intrusion into the body of pathogenic microorganisms and their subsequent multiplication.
Response 2. Thank you very much for this observation. The truth is that we have not specified where figure 1 should go. But in this new version we have proposed to the editors to put figure 1 after point 2, where we talk about the CWI in Cryptococcus. The typographical errors in figure 1 have already been corrected. And We regret this error in figure 1, it is HOG and not HO. This error has already been corrected.
I appreciate the change of figure, it is now more readable. According to the Instruction for Authors all Figures should be inserted into the main text close to their first citation, therefore I suggest to include citation in point 2.
Response 4. All section 2 refers to CWI pathway in Cryptococcus neoformans (as specified in the title of that section). When we want to point out something about Saccharomyces cerevisiae it is specified in the text but even so we have made a table to make it easier for the reader to understand.
Thank you for including the table comparing S. cerevisiae and C. neoformans. It sheds some light on interesting differences (it should be referenced in the main text as supplementary material). Based on it, I suspect that the following sentence: When the integrity of the cell wall is altered, there are specific surface receptors (Mid-2, Wsc1-3 and Mtl1), although it is not known how these receptors detect stress. Once activated, they interact with the guanine nucleotide exchange factor (GEF) Rom2 that regulate the Rho1 GTPase, which is highly regulated by many subunits.
refers to S. cerevisiae? The specification regarding C. neoformans appears in the next sentence. Such structure of this fragment is somewhat confusing. Please verify this.
Moreover, according to the comments included in the paper Inglis et al. (2014). Literature-based gene curation and proposed genetic nomenclature for Cryptococcus. Eukaryotic cell, 13(7), 878–883. https://doi.org/10.1128/EC.00083-14: “Gene names should be capitalized and italicized (e.g., ACT1), while protein names have the first letter capitalized with the rest of the letters lowercase and are not italicized (e.g., Act1).” I suggest unification of the abbreviations in the manuscript.
Author Response
POINT 1. Thank you very much for the clarification, but I still think that the word "infect" should be somewhat mitigated as it suggests intrusion into the body of pathogenic microorganisms and their subsequent multiplication.
RESPONSE 1. We thank the reviewer for this observation and therefore we have decided to modify the text and now lines 28-31 are " C.neoformans infection is very common, healthy people with intact immunity are resistant to the infection. How-ever, in those with altered immunity, this pathogen causes disease, principally cryptococcal meningoencephalitis”.
POINT 2. I appreciate the change of figure, it is now more readable. According to the Instruction for Authors all Figures should be inserted into the main text close to their first citation, therefore I suggest to include citation in point 2.
RESPONSE 2. Thank you very much for your note. We have added the figure 1 citation on line 73. Figure 1 is also cited on line 249
POINT 3. Thank you for including the table comparing S. cerevisiae and C. neoformans. It sheds some light on interesting differences (it should be referenced in the main text as supplementary material). Based on it, I suspect that the following sentence: When the integrity of the cell wall is altered, there are specific surface receptors (Mid-2, Wsc1-3 and Mtl1), although it is not known how these receptors detect stress. Once activated, they interact with the guanine nucleotide exchange factor (GEF) Rom2 that regulate the Rho1 GTPase, which is highly regulated by many subunits. refers to S. cerevisiae? The specification regarding C. neoformans appears in the next sentence. Such structure of this fragment is somewhat confusing. Please verify this.
RESPONSE 3. We are very sorry for this error and we have added lines 83-85 where we refer to table 1 of supplementary material.
In line 68, we have eliminated the names of the surface receptors (which are from S.cerevisiae) but we have kept the rest of the paragraph that refers to how the CWI pathway works in C.neoformans. We believe that this makes it less confusing for the reader
POINT 4. Moreover, according to the comments included in the paper Inglis et al. (2014). Literature-based gene curation and proposed genetic nomenclature for Cryptococcus. Eukaryotic cell, 13(7), 878–883. https://doi.org/10.1128/EC.00083-14: “Gene names should be capitalized and italicized (e.g., ACT1), while protein names have the first letter capitalized with the rest of the letters lowercase and are not italicized (e.g., Act1).” I suggest unification of the abbreviations in the manuscript.
RESPONSE 4. We have done a careful reading of the manuscript and we have modified the errors that we have founf in the text. We had used capitals and italics for genes, lowercase and italics for mutants and proteins in uppercase. But according to the reviewer's observation we have modified the typography of the proteins.
